# Missing molybdenum and the composition of the continental crust inferred from molybdenum isotopes

Yamei Tian[1,2], Feng Huang [1,3] ✉, Jifeng Xu [1,3] ✉, Jie Li[4], Yunchuan Zeng[1,3] & Alex J. McCoy-West [2,5] ✉

Accurately constraining the molybdenum (Mo) isotope composition ($\delta^{98/95}$Mo) of Earth's major reservoirs is essential for understanding its evolution. However, $\delta^{98/95}$Mo of the continental crust (CC), particularly the middle and lower crust, remains poorly constrained. Here we show the Mo isotope data for the Gangdese arc section in combination with published data from ultramafic-mafic, intermediate and felsic intrusions, representing the lower, middle and upper CC, respectively, constrain variability within the CC. Mass balance calculations using several crustal depth models generate an average $\delta^{98/95}$Mo of the bulk CC of $-0.116 \pm 0.011‰$ (2 s.d.), resolvably heavier than the bulk silicate Earth. Global scale mass balance modeling demonstrates that the Mo isotope compositions of the CC and the depleted mantle are presently near balanced. Lower crustal delamination is an additional mechanism capable of contributing to the subchondritic Mo isotope composition of the depleted mantle. Over the course of Earth's history, new crustal growth and destruction have reached dynamic equilibrium.

Earth's continental crust represents a unique geological entity within the Solar System. It fundamentally reshaped mantle geochemistry and atmospheric evolution, sustains the biosphere, and serves as a critical long-term carbon reservoir through weathering[1,2]. While its pivotal role in planetary evolution is established, the mechanisms of crustal generation, growth, recycling, and geodynamic evolution remain fundamental challenges. Zircon U-Pb geochronology constrains the earliest timing of emergence of primordial crust[3,4], while radiogenic isotope systems display spatiotemporal coupling between mantle differentiation and crustal recycling, archiving continuous crust-mantle mass exchange throughout Earth's history[5-7]. Stable isotopes in komatiites and picrites suggest rapid early crustal growth and destruction in the first billion years of Earth's history[8]. However, throughout Earth's history, crustal growth (i.e., magmatism and relamination[9,10]) and crustal

destruction (i.e., erosion, subduction and delamination[11,12]) have occurred continuously and concurrently. As a result, whether the modern crust–mantle system has reached a long-term steady-state equilibrium remains an open question.

Molybdenum (Mo) isotopes have emerged as a transformative geochemical tracer, decoding paleoredox environments[13-15] and ore genesis mechanisms[16-19]. Crucially, their dual behavior as refractory yet highly incompatible elements now extends to planetary-scale processes, from crust-mantle differentiation to deep Earth cycling[8,20-23]. However, the absence of rigorously constrained endmember compositions for Earth's principal reservoirs (e.g., mantle and crust) represents a critical barrier to: (1) tracing element recycling; and (2) quantifying fluxes between reservoirs. Characterization of these reservoirs is thus imperative for harnessing Mo's full potential to

[1]School of Earth Science and Resources, State Key Laboratory of Geological Processes and Mineral Resources, China University of Geosciences, Beijing, China. [2]IsoTropics Geochemistry Laboratory, Earth and Environmental Science, James Cook University, Townsville, QLD, Australia. [3]Frontiers Science Center for Deep-time Digital Earth, China University of Geosciences, Beijing, China. [4]State Key Laboratory of Deep Earth Processes and Resources, Guangzhou Institute of Geochemistry, Chinese Academy of Sciences, Guangzhou, China. [5]Economic Geology Research Centre, James Cook University, Townsville, QLD, Australia. ✉e-mail: fenghuang@cugb.edu.cn; jifengxu@cugb.edu.cn; alex.mccoywest@jcu.edu.au

decipher Earth's dynamic history. Chondritic meteorites, the purported building blocks of the terrestrial planets, have a relatively uniform Mo isotope composition (average $\delta^{98/95}$Mo = −0.154 ± 0.013‰, 95% s.e.; relative to NIST SRM 3134[8,24,25]), which is assumed to represent the initial composition of the bulk silicate Earth (BSE)[3,8]. The depleted mantle (DM) has maintained isotopic uniformity through geological time due to convective mixing, with a well constrained $\delta^{98/95}$Mo of −0.204 ± 0.008‰ (95% s.e.)[8]. Whereas, consensus on the Mo isotope composition of the continental crust (CC) has yet to be reached (Fig. 1). The $\delta^{98/95}$Mo of hydrothermally derived molybdenites was first used to estimate the $\delta^{98/95}$Mo of the upper continental crust (UCC)[26]. Most other estimates of $\delta^{98/95}$Mo of the UCC are based on the uppermost crust but still exhibit a wide range from +0.05‰ to +0.15‰[26–28]. In the most recent attempt, Chen et al[29]. undertook a compilation of granitoid data resulting in a largely unchanged estimate of $\delta^{98/95}$Mo for the UCC ( + 0.12 ± 0.05‰, 95% s.e.), compared to previous estimates based on either granites or subduction-related volcanic rocks[27,28]. All current data is consistent with a superchondritic Mo isotope composition for the UCC. However, systematic investigations of the middle and lower continental crust (MCC and LCC, respectively) remain scarce (Fig. 1) due to the challenges of obtaining appropriate samples and the complexities surrounding deep crustal sections[30–32]. Consequently, current estimates of the bulk continental crust (BCC) Mo isotope composition are derived through extrapolation from UCC samples that dominate existing datasets[33,34]. This methodology not only neglects MCC and LCC contributions but compounds uncertainty by propagating inherent biases in upper crustal signatures. Precise determination of Mo isotope compositions of the MCC and LCC reservoirs are therefore critical not only for constraining the BCC, but also for deciphering the geochemical evolution of Earth's exposed silicate reservoir.

Arc magmas generated above subduction zones exhibit numerous geochemical similarities to the BCC, indicating arc magmatic processes play a significant role in formation and evolution of the CC[11,35,36]. Thus, examination of a continental arc crustal section should provide better constraints on the Mo isotope composition of the CC[11,37,38]. However, the scarcity of well-preserved crustal sections limits our understanding of the Mo isotope composition of the CC at greater depths. The recently discovered arc crustal section, in the southeast Gangdese arc of southern Tibetan Plateau[35,39], is a continuous exposure of a continental arc crust (Supplementary Fig. 1), which offers an ideal opportunity to constrain the Mo isotope composition of the deeper portions of the CC. Here, we present 24 Mo isotope data obtained in this study for, ultramafic-mafic rocks (gabbros, hornblendites and pyroxene hornblendites; $\delta^{98/95}$Mo = −0.41 to +0.20‰), intermediate rocks (quartz diorites and tonalites; $\delta^{98/95}$Mo = −0.37 to +0.27‰) and

felsic rocks (granites; $\delta^{98/95}$Mo = −0.17 to −0.09‰), from Milin, Lilong and Wolong areas in the southeast Gangdese arc (Supplementary Tables 1–2). These rocks comprise common mafic (pyroxene, amphibole, biotite) or felsic (plagioclase, K-feldspar, and quartz) minerals without visible alteration[39] (Supplementary Fig. 2 and Supplementary Text). Samples from each lithology demonstrate consistent geochemical signatures (Supplementary Figs. 3–4), confirming their chemical compositions remain pristine and unaffected by secondary low-temperature metamorphism or weathering processes. In addition, we have compiled published Mo concentration and isotope data for intrusive rocks of varying compositions ($n = 306$) to help better constrain the composition of the sparsely sample lower potions of the CC (Supplementary Tables 3–4). This dataset is then used to undertake Mo isotope mass balance modeling to: (1) constrain the Mo isotope composition of the BCC based on a range of crustal depth models; and (2) undertake global reservoir scale modeling to provide valuable insights into both crust-mantle interactions and crustal recycling throughout Earth's history.

## Results and discussion
### The relationship between crustal depth, silica content and molybdenum systematics

Variations in crustal properties, such as thickness and Moho characteristics across tectonic settings, have led to the development of variable crustal depth models[37,40,41]. The proposed crustal models include: (1) a three-layer model based on mantle heat flux[37]; (2) a three-layer model with a 10-km-thick, mafic lower crust inferred from seismic data[41]; and (3) a two-layer model constrained by mantle heat flow[9]. However, all three crustal models share some fundamental similarities. The UCC is primarily composed of felsic rocks, while the deeper portions of the CC predominantly consist of ultramafic, mafic and intermediate rocks[9,37,40,41]. The deeper crust can be further subdivided into the MCC and LCC, which differ noticeably in composition. Therefore, it is important to identify which portion of the crust samples represent. In terms of lithology, both the MCC and LCC exhibit significant heterogeneity, based on exposed high-grade metamorphic rocks and crustal sections, and deep-crustal xenolith suites[37], although generally the MCC contains a higher proportion of evolved rocks (e.g., crustal sections[11,40] and granulite-facies terranes[37,42]), whilst the LCC is dominated by more mafic-ultramafic rock types (e.g., xenoliths[37]). Overall, this bulk mineralogy is consistent with the average compositions of the MCC and LCC calculated by previous studies[37,40]. A relationship between the $SiO_2$ content and P-wave velocities (Vp) of rocks, also allows the structure of the CC to be independently reconstructed, with the UCC having Vp = 5.8–6.4 km/s corresponding to felsic rocks; the MCC having

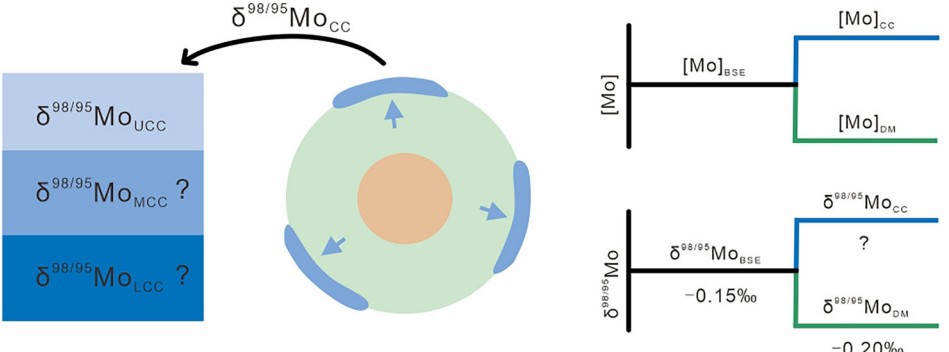

**Fig. 1 | Schematic showing Mo evolution during crust formation.** The initial $\delta^{98/95}$Mo of the bulk silicate Earth (BSE) is chondritic[14,15]. Subsequent formation of mantle and crust resulted in a depleted mantle[9] (DM) that is lighter than BSE and a continental crust (CC) that is heavier than BSE. The Mo isotope composition of the entire CC remains uncertain. The Mo isotope composition of the upper crust (UCC) has been estimated in several ways[28,29], however, the middle and lower portions (MCC and LCC) remain poorly studied. [Mo] represents the Mo concentration.

Vp = 6.4–6.8 km/s corresponding to intermediate rocks; and the LCC having Vp = 6.8–7.4 km/s corresponding to mafic rocks (excluding eclogite)[43]. The investigated Gangdese arc ultramafic–mafic rocks (Supplementary Fig. 5) have $SiO_2$ contents of 41.6–44.8 wt% (Supplementary Table 1), paleo-depths ≥31 km based on mineral crystallization pressures, and seismic velocities of 6.7–6.9 km/s[35], the intermediate rocks (53.1–57.3 wt% $SiO_2$) correspond to depths ≥ 22 km and velocities of 6.3–6.5 km/s[35], while the felsic rocks (67.8–68.8 wt% $SiO_2$) correspond to depths ca.17-21 km and velocities of 6.3 km/s[35]. These combined features indicate that the ultramafic–mafic, intermediate and felsic rocks represent the LCC, MCC and UCC, respectively.

All parts of the CC are volumetrically dominated by intrusive rocks (intrusive to extrusive volumes are typically about 10 to 1 for CC)[44,45]. Therefore, the evolution and bulk properties of the CC will be strongly controlled by magmatic intrusions. Taking this relationship between $SiO_2$ and seismic velocities a step further we can infer that ultramafic-mafic intrusions ($SiO_2$ ≤ 52 wt%) represent an analog for the LCC, while intermediate intrusions (52 wt% <$SiO_2$ < 63 wt%) broadly represent the MCC, while felsic intrusions ($SiO_2$ ≥ 63 wt%) are representative of the UCC (Supplementary Fig. 5). For example, elemental abundances of samples from the UCC, MCC and LCC exhibit clear geochemical relationships, with a strong negative correlation observed between $SiO_2$ and MgO (Supplementary Fig. 5a). Trace element compositions display similar systematic trends, for example Ni and Yb are more compatible than Rb and La during magmatic evolution, resulting in higher Rb/Ni and La/Yb ratios in the UCC (which is dominated by evolved magmas) than the other portions of the CC (Supplementary Fig. 5b, c). The investigated ultramafic–mafic, intermediate and felsic rocks herein closely match the Rudnick and Gao[37] estimates for the LCC, MCC and UCC (Supplementary Fig. 5), consistent with their formation positions in the Gangdese arc section. Overall, it appears that the compositional estimates of the different layers of the CC provided by Rudnick and Gao[37] appear the most reliable, when compared to the Gangdese arc section[35].

Determining the Mo concentrations of the different portions of the CC is crucial for understanding Mo behavior during crustal formation and differentiation processes. The composition of UCC is relatively well constrained (Supplementary Table 5) with its Mo concentration having been estimated in several ways including: (1) based on granodiorites from the Canadian Shield (Mo = 1.4 µg/g[46]); (2) based on the sedimentary rock archive (Mo = 1.5 µg/g[47,48]; Mo = 0.9–1.3 µg/g[49]); and (3) a compilation of sedimentary data and granodiorites from the Archean, Proterozoic, and Phanerozoic eras (Mo = 1.1 µg/g[37]). However, the Mo concentrations of the MCC and LCC remain poorly quantified due to limited accessibility. By following the constraints proposed by Sims et al.[50] (i.e., Mo/Ce = 0.03), Rudnick and Gao[37] determined that the Mo concentration of the LCC is ca. 0.6 µg/g, in agreement with the value obtained independently from a compilation of felsic granulite terrains and mafic xenoliths[46]. Given the sparsity of exposed crustal sections and small number of samples analyzed for Mo isotope systematics from these terranes (n = 24), to supplement the dataset here we have compiled the Mo concentration and isotope compositions of all intrusions globally subdivided into three groups: (1) $SiO_2$ ≤ 52 wt%; (2) 52 wt% <$SiO_2$ < 63 wt%; and (3) $SiO_2$ ≥ 63 wt%; to represent the different layers of the CC (i.e., LCC, MCC, UCC, respectively) as this approach will provide a more comprehensive sampling (i.e., less biased by individual outliers; n = 330) of the Mo composition of the CC. Independent of bulk composition, the Mo concentrations of intrusive rocks exhibit a wide range (Supplementary Fig. 6). Here, we take median values to most accurately represent the "true" composition of each population, given that arithmetic averages (i.e., means), can be significantly affected by anomalous outliers (cf. median values closely agree with highest relative probability values). Ultramafic-mafic intrusions with $SiO_2$ ≤ 52 wt% have median Mo concentration of 0.174 ± 0.062 µg/g (95% s.e.; n = 119; ± 0.688 µg/g, 2 s.d.),

while intermediate intrusions have median Mo = 0.649 ± 0.254 µg/g (95% s.e.; n = 52; ± 1.822 µg/g, 2 s.d.), and felsic intrusions with $SiO_2$ ≥ 63 wt% have median Mo = 0.700 ± 0.137 µg/g (95% s.e.; n = 159; ± 1.720 µg/g, 2 s.d.) (Supplementary Table 3). Clearly, Mo is significantly enriched in the UCC and MCC relative to the LCC which can be attributed to two major factors: (1) Mo strongly incompatible nature in silicate melts ($D_{Mo}$ = 0.006–0.008[51,52]); and (2) delays in sulfide saturation (due to variable $fO_2$) allowing Mo to remain in the melt phase and be transported efficiently to the UCC[8]. Despite these processes, granitoids actually exhibit significant Mo depletion. Our compilation of Mo concentrations in intrusions reveals Mo underestimation using this archive is widespread throughout the CC (Fig. 2), with Mo concentrations ca. 47% lower in the UCC, ca. 1% in the MCC and ca. 73% in the LCC than predicted previously[37,46,48]. Volumetrically dominate silicates (e.g., quartz, feldspars, biotite, and amphibole) generally contain negligible Mo at ≤ 0.2 µg/g[53–55], thus the main mineralogical hosts of Mo in the continental crust are generally weathering-resistant, titaniferous phases such as titanite, ilmenite, and magnetite[53,54,56]. Enrichment of these phases in unsampled rocks (i.e., sediments in the UCC; and eclogites in the LCC) could help explain the missing Mo in the intrusive record. Additionally, although ore-related rocks constitute a relatively small volume of the overall CC, their typically elevated Mo concentrations can significantly influence the Mo budget of the UCC and should not be overlooked. Molybdenite may be a significant Mo host in some felsic plutons[18,26,53,57] and given plutonic suites show correlations between Mo and fluid-soluble elements (e.g., Cs, Pb, As[53]), Mo loss in an aqueous magmatic vapor phase could also play a role.

Despite recording Mo depletion, the question remains can intrusions provide an accurate estimate of the Mo isotope composition of the CC. Low-temperature processes could cause significant Mo isotope fractionation, with light Mo preferentially adsorbed by weathering products (e.g., Fe-Mn oxides, clay minerals)[58,59] or organic matter[60,61], while heavy Mo is more readily transported in aqueous fluids[23,54,59]. Mineralization processes also influence Mo isotope compositions, as

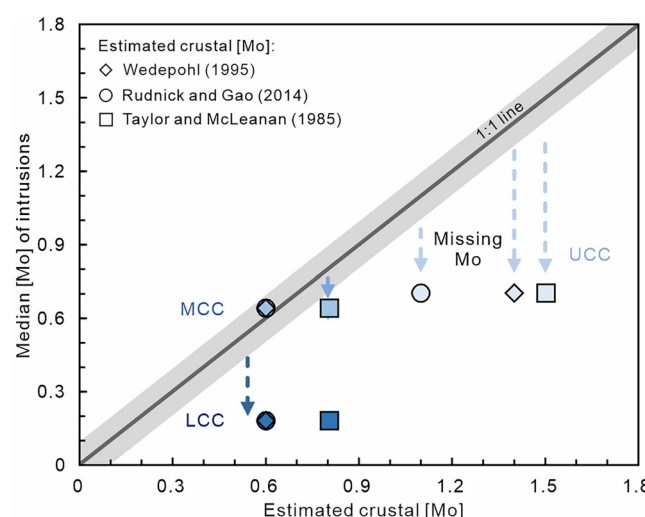

**Fig. 2 | Comparison of estimated Mo concentrations (i.e., [Mo]) of the different layers of the continental crust.** Mo concentrations of intrusions with $SiO_2$ ≤ 52 wt%, 52 wt% <$SiO_2$ > 63 wt%, and $SiO_2$ ≥ 63 wt%, represent those of the lower, middle, and upper continental crust (i.e., LCC, MCC and UCC), respectively, and were compiled herein "Median [Mo] of intrusions". "Estimated crustal [Mo]" for the lower, middle, and upper continental crust are derived from previous studies[37,46,48]. The middle and lower crust from Taylor and McLennan[48] and Wedepohl[46] have identical [Mo]. The gray shaded region indicates the Mo concentration derived from intrusions with an uncertainty margin of ±0.1 µg/g. The blue dotted arrows represent the underestimate of Mo when only considering the [Mo] of intrusions.

ore-forming hydrothermal fluids with heavier Mo isotopes can significantly alter the $\delta^{98/95}$Mo of intrusions[18,62]. To avoid any potential biases from secondary processes that may modify Mo abundance and isotope systematics, weathered and primary ore-bearing samples have been excluded[22,63]. Nevertheless, the Mo isotope composition of the compiled intrusive samples exhibits a wide range in $\delta^{98/95}$Mo from −1.12 to +0.78‰ ($n = 330$). However, the comprehensive data compilation utilized here helps to mitigate the influences of any other subordinate processes that may influence Mo isotope compositions as these outliers will have minimal impact on the derived median compositions. The median $\delta^{98/95}$Mo values (Supplementary Fig. 7 and Supplementary Table 4) are −0.172 ± 0.038‰ (95% s.e.; $n = 119$; ± 0.414‰, 2 s.d.) for the ultramafic-mafic intrusions representing the LCC, −0.130 ± 0.081‰ (95% s.e.; $n = 52$; ± 0.579‰, 2 s.d.) for intermediate intrusions analogous to the MCC, and −0.070 ± 0.060‰ (95% s.e.; $n = 159$; ± 0.762‰, 2 s.d.) for felsic intrusions representing the UCC. Firstly, for the MCC given that there is no underestimate of Mo concentrations (relative to the previously estimated Mo concentration[37,46,48]) based on intermediate intrusions (Fig. 2), we can have high confidence that our $\delta^{98/95}$Mo estimate for the MCC is indeed accurate and representative. Next, given that the LCC is volumetrically dominated by intrusive (or metamorphosed equivalents) rocks[37], despite Mo concentrations being underestimated, there is no a priori reason to suspect that our estimate of the Mo isotope composition of this reservoir should be erroneous. Lastly, when considering the UCC notably, our updated $\delta^{98/95}$Mo of the UCC is in excellent agreement with average arc magmatic rocks (−0.07 ± 0.04‰, $n = 227$, 2 s.e.[33]), but is slightly lighter than previous UCC estimates (+0.05 to +0.15‰[26–28]). To validate our UCC intrusive rock-based estimate, we also compiled Mo isotope data from sedimentary archives (diamictites, clays, and loess[59,64]) which yield a median $\delta^{98/95}$Mo of −0.070 ± 0.119‰ ($n = 45$, 95% s.e.; Supplementary Fig. 8), indistinguishable from our UCC value. This agreement demonstrates that while sediments may be enriched in Mo (explaining this missing Mo in the UCC based on intrusive rocks, Fig. 2), their $\delta^{98/95}$Mo remains identical to those of intrusive rocks making the intrusive rock record alone a robust archive for assessment of unmodified Mo isotope signatures of the UCC.

With varying $SiO_2$ intrusive rocks exhibit distinctive Mo isotope compositions, with resolvable differences observed between ultramafic-mafic-intermediate rocks and the isotopically heavier felsic intrusive rocks. Consistent with petrologic constraints, simple partial melting modeling[8] demonstrates the UCC (Supplementary Fig. 7; $\Delta^{98/95}Mo_{UCC-mantle}$ = +0.13‰) cannot be generated by direct melting of the mantle. Enrichment of the UCC in heavy $\delta^{98/95}$Mo must instead result

from intracrustal differentiation, through either the addition of isotopically heavy subduction-related fluids[22] or hydrothermal fluids[23] or the removal of isotopically light hydrous phases (biotite or amphibole)[27] into cumulates in the lower crust, consistent with complex Mo isotope behavior across crustal layers.

## Quantifying the Mo isotope composition of the continental crust

The CC exhibits a wide range of Mo concentrations and Mo isotope compositions due to the diverse nature of crustal rocks and their complex histories[24,37,53]. As demonstrated above, higher $\delta^{98/95}$Mo is related to increasing $SiO_2$ contents of intrusions (Supplementary Fig. 7), by combining this observation with the various crustal depth models (Supplementary Figs. 9–10), allows us to quantify the Mo isotope composition of the BCC. Along with the Mo isotope composition of different crustal layers derived above, the Mo concentrations and thickness of the different portions of the CC must be considered (i.e., UCC, MCC and LCC). Mass balance calculations were undertaken using three different Mo concentration datasets, the compiled intrusive Mo datasets herein (Supplementary Table 3), and previously estimated values from Taylor and McLennan[48] and Rudnick and Gao[37] for the LCC, MCC and UCC representing potential maximum and minimum values of the different crustal layers (Supplementary Table 5). Combined with the three crustal models this yields nine results that encompass all possible variations in the Mo isotope composition of the BCC (Supplementary Table 6). These results demonstrate: (1) when focusing on a single crustal depth model, the $\delta^{98/95}$Mo of the BCC calculated using Mo concentrations from intrusions alone yields a marginally heavier $\delta^{98/95}$Mo (but within uncertainty), while calculations based on the two estimated Mo concentrations[37,48] yield consistent but slightly lighter $\delta^{98/95}$Mo (Fig. 3). This demonstrates that the chosen Mo concentration of different crustal layers does not significantly impact the $\delta^{98/95}$Mo of the BCC (substantial differences will only occur with changes of Mo concentration of ≥1 µg/g); and (2) when using the same Mo concentrations, independent of the selected crustal depth model, the maximum $\delta^{98/95}$Mo variation in the BCC is ≤ 0.01‰ (Supplementary Figs. 9–10), which is negligible relative to uncertainties associated with the model and presently achievable analytical uncertainties of the input data. This demonstrates that selected crust depth model exerts minimal influence on the $\delta^{98/95}$Mo of the BCC. Ultimately, the Mo isotope composition of the BCC calculated herein using reliable Mo concentration estimates[37,48] and the three crustal depth models[37,40,41] are indistinguishable (Fig. 3) and provide an average $\delta^{98/95}$Mo for the BCC of −0.116 ± 0.011‰ (2 s.d.; $n = 6$; Fig. 3), validating the superchondritic $\delta^{98/95}$Mo of the CC as suggested previously[28,29,34].

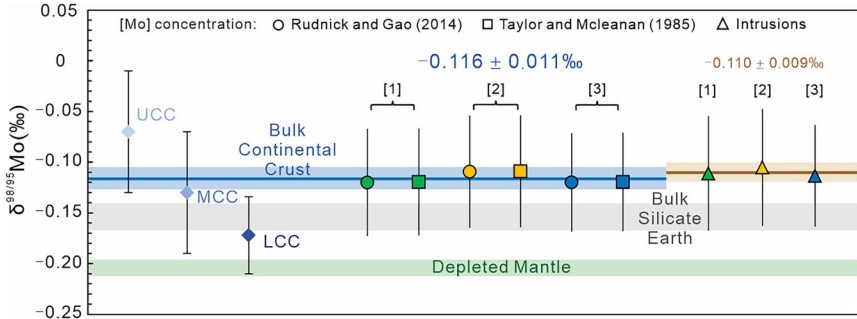

**Fig. 3 | Mo isotope composition (i.e., $\delta^{98/95}$Mo) of the bulk continental crust calculated using different Mo concentrations for the lower, middle and upper continental crust (i.e., LCC, MCC and UCC) and applied to three different crustal models.** The diamonds represent the $\delta^{98/95}$Mo of UCC, MCC and LCC, respectively. [1], [2] and [3] represent crustal section models from Rudnick and Gao[37], Huang et al.[41] and Hacker et al.[40], respectively. The blue line, along with its corresponding blue bars, represent the average and range of $\delta^{98/95}$Mo (−0.116 ± 0.011‰, 2 s.d.) calculated from previously estimated crustal [Mo][37,48]. The brown line, along with its corresponding brown bar, represent the average and range of $\delta^{98/95}$Mo (−0.110 ± 0.009‰, 2 s.d.) calculated from the intrusive rock compilation. The comparative green and gray bars represent the $\delta^{98/95}$Mo of the depleted mantle (−0.204 ± 0.008‰[8]) and bulk silicate Earth (i.e., chondrites; −0.154 ± 0.013‰[8],). Uncertainties on the $\delta^{98/95}$Mo of the UCC, MCC and LCC and bulk continental crust are 95% standard error.

## Evaluating the Mo isotope composition of the continental crust

To determine whether the Mo isotope composition of the BCC determined herein is realistic, we have undertaken reservoir scale mass balance modeling to validate its robustness. McCoy-West et al.[8] proposed that the only remaining global-scale mechanism (following planetary accretion and core formation) that could modify Earth's Mo isotope budget and account for Earth's superchondritic crust and subchondritic depleted mantle was crust extraction. Subsequently, Hin et al.[65] proposed that subducted oceanic crust, which has isotopically light $\delta^{98/95}$Mo[66] and high Ce/Pb[67], mixed back into the upper mantle, also could explain the subchondritic nature of the mantle. Ultimately, both hypotheses require that Earth's superchondritic crust and subchondritic mantle for Mo result from geological processes in the crust-mantle system. Therefore, here we assume a model involving a complementary crust and depleted mantle, and have calculated the crustal volume that is required to reconcile the Mo concentration and $\delta^{98/95}$Mo of the crust using the mass balance equation[8]:

$$m_c = \frac{m_{DM} \times [Mo]_{DM} \times (\delta^{98/95}_{BSE} - \delta^{98/95}_{DM})}{[Mo]_C \times (\delta^{98/95}_C - \delta^{98/95}_{BSE})}$$

where $m_i$, $[Mo]_i$ and $\delta^{98/95}_i$ represent the mass, Mo concentration and Mo isotope composition, respectively, of the various reservoirs. Noting the mass balance modeling undertaken here does not reflect the instantaneous removal of melts from the mantle, but rather the time-integrated isolation of the crust from the mantle over Earth's history. In this calculation, the crust (C) is the sum of both the continental crust (CC) and oceanic crust (OC) extracted from Earth's mantle at any time. Despite the smaller volume of oceanic crust (-2.4 × 10^9 km^3,[3,68],) compared to continental crust (7.1 × 10^9 – 8.4 × 10^9 km^3, Supplementary Table 7), two models are employed here to incorporate all possibilities: one model includes only the CC, while another incorporates both the CC and OC. The composition of the CC used is based on the recommended $\delta^{98/95}$Mo of the BCC presented here, with parameters for the other reservoirs presented in Supplementary Table 7. To represent the OC, the Mo concentration and $\delta^{98/95}$Mo of mid-ocean ridge basalt (MORB) have been compiled from literature data[8,20,24,65,69,70] to obtain the median Mo concentration of 0.420 ± 0.101 μg/g (95% s.e.; $n = 99$; ± 1.01 μg/g, 2 s.d.) and $\delta^{98/95}$Mo of −0.180 ± 0.028‰ (95% s.e.; $n = 99$; ± 0.278‰, 2 s.d.) (Supplementary Fig. 11).

Extraction of the continental crust did not cause wholesale depletion of the entire primitive mantle, with most models based on elemental and radiogenic isotope ratios requiring only 25–50% partial melting of the depleted mantle[7,71,72]. With just the upper portion of the mantle (e.g., 660 km; equivalent of 30% mantle depletion) considered most likely to have been modified based on geophysical evidence[73,74]. When considering solely the CC and independent of the chosen crustal depth model[37,40,41], this modeling shows several important results based on using a conservative minimum estimate of 25% mantle depletion (i.e., $M_{DM}/M_{BSE} = 0.25$): (1) when using the Mo concentration of intrusive rocks, the calculated crustal volume significantly exceeds unity with $V_C/V_{PC} = 2.43$–3.01 (Fig. 4a; $V_C/V_{PC}$ represents volume of crust ($V_C$) relative to the volume of present-day crust ($V_{PC}$)). As discussed above, Mo concentrations based solely on intrusions do not accurately represent the CC and therefore this discrepancy is not surprising; (2) when using the previously estimated Mo concentrations, the required volume of the CC aligns closely with the modern volume of CC with $V_C/V_{PC} = 1.76$–2.07 and 1.31–1.54 based on values from Rudnick and Gao[37] and Tayor and McLennan[48], respectively. This result suggests the concentration estimates[48] provide a robust representation of the concentrations of the different layers of the CC for Mo and therefore models using these parameters are preferred here; (3) the excellent agreement between the calculated volume of crust and the present-day volume

of CC (i.e., $V_C/V_{PC} \approx 1$) confirms that the average composition derived herein based on the Mo isotope composition of intrusions must provide a valid representation of the BCC. When modeling incorporates both the CC and OC, the calculated crustal volumes increase slightly to $V_C/V_{PC} = 1.96$–2.32[37] and $V_C/V_{PC} = 1.41$–1.65[48] (Fig. 4b). This model produces only marginally higher $V_C/V_{PC}$ (ca. 10%) indicating that the OC has limited effect on the overall mass balance due to its lower Mo concentration and minimal difference in $\delta^{98/95}$Mo from the DM.

Our study confirms the heavier Mo isotope compositions of different layers of the CC with increasing silica content and decreasing depth. Given the missing Mo in the intrusive record of the LCC, this observation also offers an alternative perspective on the origin of the subchondritic $\delta^{98/95}$Mo of the mantle. During formation of the CC, delamination plays a critical role in explaining the average andesitic composition of the BCC[11,38]. In addition to Mo isotope fractionation during large-scale crustal extraction[8] and subduction dehydration[20], which can cause Mo isotope fractionation, delamination may also facilitate the recycling of ultramafic intrusions back into the mantle, contributing to the subchondritic $\delta^{98/95}$Mo of depleted mantle. A likely candidate for these recycled crustal components is eclogites which contain a significant proportion of rutile (1.5 vol.%[75]), as rutiles are characterized by elevated Mo concentrations and light $\delta^{98/95}$Mo (2.44-5.30 μg/g and average −0.77‰[20]). When the LCC undergoes thickening, the densest components (e.g., eclogites) will be preferentially delaminated into the mantle. Removal of this isotopically light eclogite-dominated component would contribute to the isotopically light composition of the depleted mantle and correspondingly, heavier BCC.

More broadly when considering Mo isotopic equilibrium between the CC and DM, we can assess the temporal evolution of the crust-mantle system. Extensive extraction and recycling of the crust (2.5 times to 3.8 times of present volume of CC) have been suggested to have occurred by the late Archean[8]. However, for the modern Earth the Mo isotope compositions of the CC and the DM are presently closer to unity ($V_C/V_{PC}$ around 1), than on the early Earth. Therefore, between 3.5 Ga and today extensive crust removal ($V_C/V_{PC} > 1$) was necessary to achieve the current near-balanced state. A substantial portion of this mismatch can be attributed to early crustal overturn[44,76]. On the modern Earth, the processes of delamination[37] and subduction[32] are the primary mechanisms responsible for the removal of the CC to the mantle. Ultimately, independent of the mechanisms at play, from a Mo perspective, net crustal growth and recycling back into the mantle have reached secular dynamic equilibrium over the course of Earth's history.

## Methods

### Major and trace elements

Twenty-four fresh whole-rock samples, fresh surfaces, were sliced into small pieces and ultrasonically cleaned with distilled water. After drying, the samples were crushed to particles smaller than 1 mm and subsequently ground to a fine powder (<200 mesh) using an agate mortar. The resulting powders were used for geochemical analyses. Loss on ignition (LOI) was determined by heating approximately 0.5 g of the sample powder in porcelain crucibles at 1000 °C for 1.5 h. Major element concentrations were analyzed using X-ray fluorescence spectrometry on fused glass beads, employing a PANalytical AXIOS Minerals spectrometer at the Rock−Mineral Preparation and Analysis Laboratory, Institute of Geology and Geophysics, Chinese Academy of Sciences, Beijing, China. The analytical precision for major elements was better than ±3%. For trace element analyses, 50 mg aliquots of the sample powders were digested with a mixture of $HNO_3$ and HF acids in high-pressure Teflon vessels for 30 h to ensure complete dissolution. Trace element concentrations were determined by the method of inductively coupled plasma mass spectrometer (ICP-MS) PlasmaQuant MS Elite (Analytik Jena, Germany), at Institute of Geochemistry,

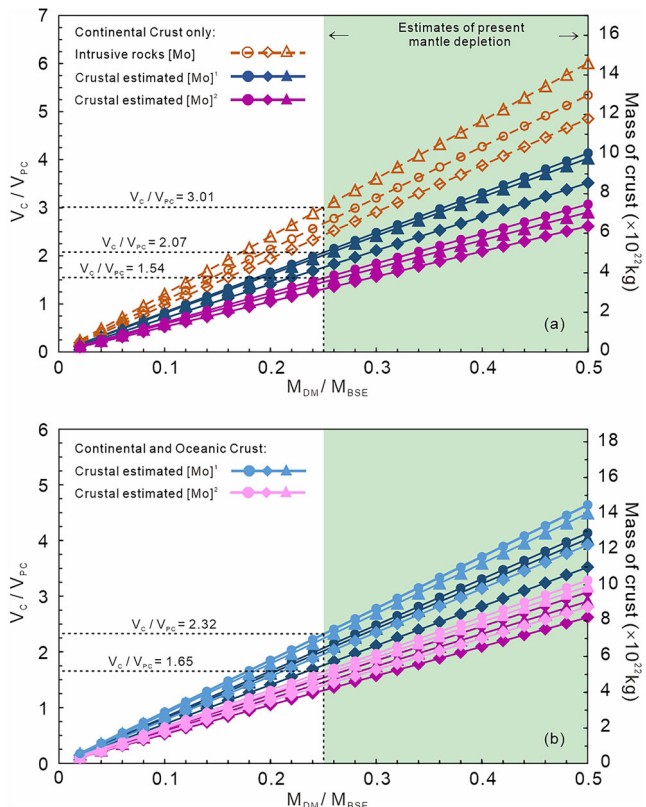

**Fig. 4 | Mass balance model assessing crust-mantle differentiation, to evaluate the validity of the our estimated δ⁹⁸/⁹⁵Mo of the bulk continental crust.** Crustal mass can subsequently be translated into a corresponding volume of crust ($V_C$) relative to the volume of present crust ($V_{PC}$), with variations depending on the fraction of the bulk silicate Earth (BSE) that has experienced melt depletion ($M_{DM}$/$M_{BSE}$). Two models were tested: (1) only continental crust (**a**) and; (2) considering both continental and oceanic crust (**b**). Previously estimated crustal [Mo][1] and [Mo][2] are from Rudnick and Gao[37] and Taylor and McLennan[48], respectively. In (**a**), the dotted lines represent those calculate using only the compiled [Mo] of intrusive rocks. Shapes represent different crustal depth models from Rudnick and Gao[37] (circle), Huang et al.[41] (diamond) and Hacker et al.[40] (triangle). In (**b**), the y-axis labeled "Mass of crust" is applicable only to the results of the continental and oceanic crust model, continental crust only model (from a; darker colors) are shown for comparison.

Chinese Academy of Sciences, Guiyang, China. For the vast majority of trace elements, ICP-MS achieves a relative standard deviation of better than 10% in reproducibility tests[77].

## Mo isotopes

Chemical separation of Mo was carried out at Guizhou Tongwei Analytical Technology Company Limited, Guizhou, China, following the procedures outlined by Li et al.[78]. All sample preparation and digestion work were undertaken in a Class 1000 clean laboratory equipped with Class 100 laminar-flow exhaust hoods. Sample powders (0.2–3.0 g) and an appropriate amount of ⁹⁷Mo-¹⁰⁰Mo double-spike solution were weighed accurately in 15 mL PFA beakers. The double-spike solution was added to account for Mo isotopic fractionation during column separation and to correct the instrumental mass bias during mass spectrometric analyses[79,80].

Samples were dissolved in HF + HNO₃ volumes appropriate for the sample size: for ~3 g samples, ~12 ml of a 2:1 mixture of 22 M HF and 14 M HNO₃ was used; for ~0.5 g samples, ~6 ml of a 2:1 mixture of 22 M HF and 14 M HNO₃ was used. Then the closed beakers were heated at 150 °C for at least 3 d. To aid sample digestion, the mixture was agitated occasionally for ~5 min in an ultrasonic bath. After digestion and

dryness at 120 °C, the samples were dissolved in 1–3 ml of concentrated HCl and again evaporated to dryness. The residue was redissolved in 2 ml of a mixture of 0.1 M HF and 1 M HCl, at which point it was ready for column separation. The IAPSO Atlantic seawater sample (ca. 10 ml) was acidified with 1 ml of concentrated HCl, and the ⁹⁷Mo-¹⁰⁰Mo double spike was then added. The spiked and acidified seawater solution was then evaporated to dryness before being redissolved in 4 ml of 0.1 M HF and 1 M HCl.

Prior to use, the N-benzoyl-N-phenylhydroxylamine (BPHA) resin column was washed sequentially with 6 ml of 6 M HF and 1 M HCl, followed by 4 ml of Milli-Q water. Before sample loading, the column was conditioned with 2 ml of a mixed solution containing 0.1 M HF and 1 M HCl. The chemical separation procedure, along with previously established methods are outlined in Li et al.[78]. A 2 ml aliquot of the sample solution was loaded onto the column, after which the resin was washed with 8 ml of 0.1 M HF and 1 M HCl. This washing step effectively removed matrix and interfering elements (Mn, Fe, Ni, Cu, Zn, Zr, Ru, etc.) from the samples. The adsorbed Mo was then eluted using 8 ml of 6 M HF and 1 M HCl. The eluate containing Mo was collected in 15 ml PFA vials and evaporated to dryness on a hot plate at 120 °C. To decompose any residual organic matter, three drops of concentrated HNO₃ and H₂O₂ were added to the dried Mo deposit. Subsequently, 1 ml of 3 vol% HNO₃ was introduced to dissolve the residue, and the resulting solution was prepared for Mo isotope ratio measurement by Multi-Collector (MC)-ICP-MS. It is noteworthy that the concentrations of Mn, Fe, Cr, Ni, Zr and Se in the final Mo solution were very low (a few ng/ml), and Ru was not detected in the analyte solution by ICP-MS after separation.

Molybdenum isotope ratios were determined on a Thermo-Fisher Scientific Neptune Plus MC-ICP-MS at Guangzhou Institute of Geochemistry, Chinese Academy of Sciences, Guangzhou, China. This instrument was equipped with eight moveable Faraday collectors and one fixed central collector, which were linked to amplifiers with 10¹¹ Ω resistors.

Samples and calibrator solutions were introduced into the plasma through an Aridus II desolvating system (CETAC Technologies, Omaha, USA). The system removes solvent by passing the aerosol through a membrane heated to 160 °C, thereby delivering a 'dry' sample to the plasma. This process enhanced signal sensitivity by a factor of 5–10 compared to conventional wet plasma introduction. Under these conditions, a typical Mo sensitivity of 180–200 V/µg ml⁻¹ was achieved. The Faraday collector configuration on the Neptune Plus MC-ICP-MS was set as follows: L4: ⁹¹Zr, L3: ⁹²Mo, L2: ⁹⁴Mo, L1: ⁹⁵Mo, C: ⁹⁶Mo, H1: ⁹⁷Mo, H2: ⁹⁸Mo, H3: ⁹⁹Ru and H4: ¹⁰⁰Mo. Data for all samples and standards were acquired in six blocks of ten ratios. The resulting sixty ratios were then processed using a one-pass 2σ outlier rejection criterion, from which the mean value and standard error (s. e.) were calculated. Following each analysis, the introduction system was rinsed sequentially for 10 min with 3 vol% HNO₃, 0.2 M HF plus 3 vol% HNO₃ and 3 vol% HNO₃. This cleaning procedure typically reduced the residual Mo signal to below 1 mV, a level negligible compared to the analyte signals measured during analysis.

Correction for mass fractionation during chemical separation as well as during mass spectrometry followed the double-spike deconvolution methods described by Siebert et al.[18]. Molybdenum isotopic ratios of all samples were normalized to those of the NIST SRM 3134 Mo reference material and expressed in conventional δ notation:

$$\delta^{98/95}\text{Mo}(‰) = [^{98/95}\text{Mo}_{\text{sample}} / ^{98/95}\text{Mo}_{\text{NISTSRM3134}} - 1] \times 1000 \quad (1)$$

During the measurement, the total procedural blanks for Mo were 0.39 ± 0.37 ng (2 s.d.; $n$ = 7). IAPSO Atlantic seawater and the USGS rock standards AGV-2 and W-2a were also analyzed and yielded δ⁹⁸/⁹⁵Mo values of +2.06 ± 0.03‰, −0.15 ± 0.03‰, −0.04 ± 0.06‰, respectively

(Supplementary Table 2). The $\delta^{98/95}$Mo values of these standards are consistent with those from previous studies[20,81,82]. Mo concentrations were determined using the isotope dilution method. IAPSO Atlantic seawater and the rock standards AGV-2 and W-2a yielded Mo concentrations of 10 ng/g, 1.99 µg/g, 0.44 µg/g, respectively (Supplementary Table 2).

## Bulk continental crustal mass balance modeling

We calculated the Mo isotope composition of the bulk continental crust using a layer-by-layer mass-balance model. The continental crust was divided into upper, middle, and lower crustal layers, each characterized by Mo concentrations and $\delta^{98/95}$Mo values (Supplementary Tables 3–5), together with representative thickness proportions (Supplementary Figs. 9–10) and densities (Supplementary Table 5). For each layer, the mass was estimated by multiplying its thickness, density, and area. This value was then combined with the average Mo concentration to calculate the Mo mass contained in that layer. The equation is given as follows:

$$m = h \times a \times \rho \tag{2}$$

where $m$, $h$, $a$, and $\rho$ are the mass, thickness, crustal area, and density of each crustal layer (UCC, MCC and LCC), respectively (Supplementary Table 5).

The Mo masses of all three layers were summed to obtain the total Mo budget of the crust. The relative contribution of Mo from each layer (i.e., the Mo mass fraction of that layer relative to the total) was used as the weighting factor for calculating the average Mo concentration and isotope composition of the CC. The equation is given as follows:

$$[Mo]_{CC} = \frac{m_{UCC} \times [Mo]_{UCC} + m_{MCC} \times [Mo]_{MCC} + m_{LCC} \times [Mo]_{LCC}}{m_{UCC} + m_{MCC} + m_{LCC}} \tag{3}$$

where [Mo] is the Mo concentration of the CC, UCC, MCC and LCC. $m$ is the mass of the UCC, MCC and LCC.

$$\delta^{98/95}_{CC} = \frac{m_{UCC} \times [Mo]_{UCC} \times \delta^{98/95}_{UCC} + m_{MCC} \times [Mo]_{MCC} \times \delta^{98/95}_{MCC} + m_{LCC} \times [Mo]_{LCC} \times \delta^{98/95}_{LCC}}{m_{UCC} \times [Mo]_{UCC} + m_{MCC} \times [Mo]_{MCC} + m_{LCC} \times [Mo]_{LCC}} \tag{4}$$

where $\delta^{98/95}$ is the Mo isotope composition of the CC, UCC, MCC and LCC. The other terms denote similar meanings as in Eq. (3).

This approach ensures that both the chemical characteristics of each crustal level and their relative volumetric and density-controlled mass proportions are considered. In addition, the model framework allows incorporation of uncertainties in isotope composition, thereby yielding a statistically robust estimate the bulk continental crust $\delta^{98/95}$Mo.

## Global scale mass balance modeling

The distribution of $\delta^{98/95}$Mo between the depleted mantle and the crust after differentiation can be estimated using isotopic and elemental mass balance[34]. Previous studies suggest that 25–50% of the whole mantle has been depleted at present day[7,8,71,72]. By combining the Mo concentration and $\delta^{98/95}$Mo of the bulk silicate Earth with those of the depleted mantle, considering the range of present-day estimates of mantle depletion, and integrating the Mo concentration and $\delta^{98/95}$Mo of the continental crust estimated in this study, we can assess the reasonableness of the present-day continental crust Mo concentration and $\delta^{98/95}$Mo estimated herein.

The equations presented here are identical to those used previously in McCoy-West et al.[8]. Here, we consider that Mo of a portion of the BSE has been accessed for crust formation and is distributed among two reservoirs: a depleted mantle (DM) and Earth's crust (C).

The isotopic mass balance can be written as follows:

$$m_{BSE} \times [Mo]_{BSE} \times \delta^{98/95}_{BSE} = m_C \times [Mo]_C \times \delta^{98/95}_C + m_{DM} \times [Mo]_{DM} \times \delta^{98/95}_{DM} \tag{5}$$

where $m$ is the mass, [Mo] is the Mo concentration and $\delta^{98/95}$ is the Mo isotope composition of the various reservoirs (BSE, DM and C). See Fig. 1.

The pure elemental mass balance is:

$$m_{BSE} \times [Mo]_{BSE} = m_C \times [Mo]_C + m_{DM} \times [Mo]_{DM} \tag{6}$$

where the terms denote similar meanings as in Eq. (5). By combining Eqs. (5) and (6), we can obtain the following equation:

$$m_C = \frac{m_{DM} \times [Mo]_{DM} \times (\delta^{98/95}_{BSE} - \delta^{98/95}_{DM})}{[Mo]_C \times (\delta^{98/95}_C - \delta^{98/95}_{BSE})} \tag{7}$$

This allows us to calculate the mass of crust generated assuming various amounts of depletion of the mantle reservoir (Fig. 4).

The volume of this crust can then be calculated using the following:

$$V_{crust} = m_{crust} / \rho_{crust} \tag{8}$$

where $\rho_{crust}$ represents the density of the crust in different crustal models, and $V_{crust}$ represents the volume of the crust.

## Data availability

The data generated or analysed during this study are included in this published article and its Supplementary Information file.

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

## Acknowledgements

We thank Mingjian Li and Song Zhang for their assistance in the field work and appreciate Jing Hu, Liying Zhang and Le Zhang for their help in geochemical analyses. This research was supported by the National Key Research and Development Project of China (2020YFA0714800), the National Natural Science Foundation of China (42121002) to J.X., the National Natural Science Foundation of China (42373045), the Fundamental Research Funds for the Central Universities of China (379202403) to F.H. and the Fundamental Research Funds for the Central Universities of China (2652023001) to J.X., Y.T. is supported by a scholarship from the China Scholarship Council (202306400068). A.J.M.-W. is supported by Australian Research Council grant DE210101395.

## Author contributions

F.H. and J.X. designed the project and collected samples with Y.Z., F.H., J.L. and Y.T. conducted the experiments and chemical analyses. Y.T. and A.J.M.-W. performed mass balance modeling, Y.T. wrote the original draft with input from F.H. and A.J.M.-W. All co-authors contributed to editing of the manuscript.

## Competing interests

The authors declare no competing interests.
