## [Transparent Peer review file · Nature Communications]

Missing molybdenum and the composition of the continental crust inferred from molybdenum isotopes

Corresponding Author: Dr Feng Huang

Version 0:

Reviewer comments:

Reviewer #1

(Remarks to the Author)

This manuscript 'Missing molybdenum and the composition of the continental crust inferred from molybdenum isotopes' by Tian et al. have measured Mo isotope data in rocks from Gangdese arc which preserves a continuous section of lower-middle-upper crustal rocks. The authors have used their data combined with further data compilation to come up with Mo concentration and Mo isotope ratio of lower, middle, upper crust and bulk continental crust. This work is very timely and has great relevance to the field of research in stable isotope geochemistry, crust-mantle evolution. However, in the current state, the manuscript has some critical shortcoming which must be rectified before the manuscript is published. The authors have only measured the rocks from the middle crustal section of the arc and there is significant potential if rocks from upper and lower crustal section are investigated and are incorporated in this study. Two major concerns that I have from this manuscript are presentation of isotopic data and calculation of main input parameters controlling the isotopic values of different crustal sections. Basic details about isotope measurements and measured values of standards are missing which hinders in assessing the quality of the data. In the present form, the work cannot be reproduced. The authors are requested to look at the line comments where it is mentioned in detail about why some of the input parameters are likely erroneous, particularly the $\delta^{98/95}\text{Mo}$ of the UCC (Upper Continental Crust). The Significant changes are therefore needed which will require re-running the models, which may change the proposed value of the BCC (Bulk Continental Crust).

Line Comments:

Line 42 - 43: Significant amount of work using Mo isotopes has been done in the recent years on different settings to understand the crust mantle evolution through geological time. Do also refer to such more recent works from the past 2-3 years, which cover these aspects of crust-mantle evolution and interaction to strengthen the point, that is, better constraints are needed for Mo isotope composition of different geochemical reservoirs.

Line 76 - 78: I was unable to find the details of Mo isotope measurements in the manuscript and in the supplement. The authors have only reported the measured Mo isotope ratios. Only with this information it is hard to assess the quality of the data as no sample preparation, column chemistry and measurement protocol has been given. There is no mention of Ru correction and procedural blank. There is also no report of the isotope spike, solution and whole rock standard(s) that have been used (if so) during the measurement. Without the measured Mo concentration and Mo isotope data of the solution and whole rock standards, and details of the measurement procedure, one cannot assess the quality of the Mo isotope data of the samples reported here. Therefore, the authors must include a section dedicated to the details of Mo isotope measurement.

Line 78: The authors need to explain the mineralogy of the studied samples to better classify them and rule out alteration. Even basic REE diagrams have not been shown. Given the implications the authors are trying to derive from these few samples, much detailed discussion is needed about them.

Line 79: The supplementary table S1 that is mentioned here has whole-rock data. If the whole rock data was measured by the authors as part of this study, details of how the whole rock data was measured must be given. This is one more instance where basic information of data gathering is totally absent. Also, the isotope data (Sr, Nd) that has been reported about these samples, were they measured by the authors? If so, then details of those measurements also must be given. If not, then the source of these data must be cited.

Even though the authors mention the Sr and Nd data in table, this data has not been used anywhere in the manuscript, and I seem to not find any discussion related to this data. It might be helpful for the authors to discuss the Sr and Nd isotope data in the context of mid crustal evolution.

Line 125 -187 In this portion of the manuscript the authors have gone into details about the possible Mo concentration and $\delta^{98/95}\text{Mo}$ of the UCC, MCC and LCC. These are the fundamental input parameters for the newly proposed isotope

composition of BCC. However, the way they have chosen the [Mo] and $\delta^{98/95}\text{Mo}$ raises significant questions. The authors have indicated the effect of the hydrothermal fluid activity on Mo isotope fractionation and have mentioned they have taken steps to exclude samples with ore-bearing fluids. If we look at the citations, they have used to calculate the [Mo] and $\delta^{98/95}\text{Mo}$ of UCC, all of them (Xue et al., Shen et al., Kaufmann et al., Fan et al., Greber et al.) except one (Yang et al. and some fresh igneous rocks in Graber et al.) are rocks with hydrothermal alteration and/or ore-fluid related process. Also bracketing rocks from same hydrothermally altered/ore-fluid related processes based on their SiO_2 content to represent different portions of the crust completely ignores the local geological effects on the [Mo] and $\delta^{98/95}\text{Mo}$. It is therefore obvious that the [Mo] and $\delta^{98/95}\text{Mo}$ derived from these rocks will be erroneous if implemented for UCC and in some cases MCC and LCC, a conclusion they themselves have come to later for [Mo]. But surprisingly even though the authors accept the [Mo] calculated by them through the compilation is missing [Mo], they have accepted the calculated $\delta^{98/95}\text{Mo}$ from the same samples without any deliberation. $\delta^{98/95}\text{Mo}$ of UCC that the authors are using in the model is $0.000 \pm 0.053\%$, which significantly differs from any previous estimates and the value commonly used (around 0.15; McCoy-West et al. and references therein). The input parameters for the [Mo] and $\delta^{98/95}\text{Mo}$ need to be changed after careful consideration and the subsequent models must be adjusted accordingly, to give new estimates on BCC $\delta^{98/95}\text{Mo}$.

Another aspect of this study which can significantly improve and should be implemented in the manuscript is measuring the [Mo] and $\delta^{98/95}\text{Mo}$ of the rocks that represent the UCC and LCC portions of the Gandese arc. This will give the authors [Mo] and $\delta^{98/95}\text{Mo}$ of rocks which are from different portions of the crust, which will significantly diminish the need of using $\delta^{98/95}\text{Mo}$ from compiled rock data which may not actually represent the different crustal sections. The Mo isotope data from surface sediments (sediments being a good representation of the exposed crust) should also be considered during estimation of the UCC composition.

Line 217: Model presented here must be readjusted according to the modified BCC value.

Reviewer #2

(Remarks to the Author)

Over the last ten years, the mass dependent Mo isotope system has shown to be a promising tool in the context of understanding the geodynamic coupling of geochemical reservoirs, such as the mantle and the crust. While still limited in number, Mo isotope data for mid-ocean ridge basalts (MORB) and meteorites have been used to establish bulk compositions for the depleted mantle and the chondritic reservoir, respectively, although one will have to admit that the total variations observed for other geochemical tracers for samples from these reservoirs indicate that the full breath of variations might not have been appreciated yet. Worse still, the Mo isotope composition of the lower mantle remains poorly constrained, as it is only accessible through the analyses of ocean island basalts, for which there is a limited number of Mo isotope data. Things become even more patchy when addressing the bulk composition of the continental crust. For one this is because of its well-known lithological heterogeneity, but also because many of the rock types building up the crust are show Mo isotope variations much larger (on average by a factor of 10) than those observed for magmatic rocks. The latter involve sediments and ore-forming (hydrothermal) deposits.

The main thrust of this contribution is to model the Mo isotopic composition of the bulk continental crust. This is done by applying a standard mass balance approach as has been done many times before using other isotopic and geochemical systems. This seems to be an obvious thing to do and I am sure that many groups do have the same approach in mind would it not be for the rather limited database such an exercise would rely on. In this context, the authors provide 13 more data points for crustal samples bringing the total number of data points for samples of the continental crust to ca. 250.

While the effort of the authors is commendable and a step in the right direction in terms of establishing the Mo isotopes system as a geodynamic tracer, I would like to point out a few issues. First, the role of sediments and ore-related rocks appears to be not properly acknowledged in this study. At several points, the author use "converging values" as an argument for validating their results. For instance, they argue that a perceived normal distribution of Mo concentrations in intrusive crustal rocks justifies their choice of specific average Mo concentrations for the lower, middle and upper continental crust, although trace element concentrations in rocks cannot be normally distributed (e.g. also see Fig. S4, that would suggest negative Mo concentrations in some crustal rocks if this would be the case). Also, they somewhat dismiss the effect of sediments and ore-related rocks on the Mo isotope budget of the continental crust, arguing that because different models converge at a similar value somewhat verifies this view. At last, the fact that this study is based on merely 250 data points for a reservoir that shows the largest isotopic and geochemical variations of all geochemical reservoirs puts a large uncertainty on their conclusions. In this context, it does not help that the authors freely switch between using 2 sigma and 2 sigma mean uncertainty estimates in their models and conclusions.

Besides these more "technical" issues, the author's undertaking might be primarily targeted towards researchers working on the Mo isotope system or geochemists with a general interest in the field of mass-dependent isotope variations in magmatic rocks. Only at the very end do the authors put their conclusions in the wider context of geochemically balancing the continental crust and the depleted mantle (a concept that is also widely accepted within the isotope geochemistry community). However, even this outlook might have to be weighted in the context of the large uncertainty due to limited data set.

Overall, although the study provides only a very limited set of new Mo isotope data and the fact, that the underlying approach is not entirely novel, I still think that this work should be published as it would be of primary interest to an expert geochemical

audience concerned with either improving the general knowledge of the bulk composition of the continental crust or an interest in fully establishing the Mo isotope system as a tracer in isotope geochemistry.

More specific comments:

Introduction

This provides an historic overview of the development of the Mo isotope system to target Earth Science-related questions. This has long been the standard way of introducing the system in publications, i.e. starting with low-temperature applications and ore geology before homing in on their application to magmatic processes. I find this quite old-fashioned given that there is now ample literature on the application of Mo isotope to geodynamic questions. I would suggest the authors re-work this part accordingly.

Line 52: I do not agree with this notion. As indicated by the authors, the uncertainty of the average composition of chondrites is small (accordingly given as 2 standard error) but this does not mean that the Mo isotopic composition of the chondritic reservoir is homogeneous. Not appreciating this here, I fear that this is (already) a major obstacle in what will follow in the context of the overall aim of this study.

Line 57: Please provide context. Unchanged compared to what?

Line 63: This is misleading. Please rephrase. I presume the authors mean that the estimated $d_{98}\text{Mo}$ value for bulk CC is biased towards upper CC values due to sampling bias, not that the databases for $d_{98}\text{Mo}$ in upper CC rocks are biased (e.g. because of analytical artefacts)?

Lines 72 to 80: Even though the Gangdese arc might have been recently discovered, I believe that this is more of a regional interest. Quite on the contrary, I think that adding the 13 data points for the samples from the Gangdese arc (here and in the next section – i.e. lines 87 to 187) in this context is misleading. This small number of data points do provide some additional information but, truly, not much and the whole project is built on literature data in the first place. As such, I would re-write these parts of the manuscript with a focus on the literature data only and, accordingly, move focus away from the (small number) of additional new data.

Line 119: Using a highly incompatible/highly compatible element pair (such as Rb/Ni) is an odd ratio. What is the underlying reason for this? Why not use Eu/Eu^* , which would be the obvious choice when discriminating between the UCC and the LCC?

Lines 120 to 125: I don't think one can use the trace element systematic for individual crustal samples to attribute them to a specific position within the crust ("... confirming their formation in the MCC"). I find this a very long stretch. Please provide further substantiation.

Lines 135 to 149: I am still not convinced that this simple "chemo-stratographic" approach is valid. However, there is an easy test that I would ask the authors to carry out in a revised version. If the Mo concentration of LCC, MCC, and UCC can be estimated by querying the SiO_2 content of analysed rocks, then the identical approach should work with other incompatible elements. This means that taking the same samples and applying the SiO_2 filter, the so-calculated average concentrations for all other incompatible trace elements should match the estimates of the same elements in the LCC, MCC and UCC in the literature (e.g. Rudnick and Gao).

Line 152: Comma missing after "this"?

Lines 174 to 178: Is this used (later?) to validate the approach? If so, I would strongly disagree. Just because the data shows a normal distribution cannot be used to argue that the data is representative for the average composition of different crustal compositions. Something that "looks right" does not a priori imply it is right. If the authors think otherwise, they need to provide a comprehensive argumentation to do so.

I would argue that sedimentary rocks as well as ore-related lithologies are a very important lithology when estimating the bulk composition of crustal reservoirs. The fact that the authors approach underestimates the Mo concentrations in the LCC and UCC compared to established estimates from the literature is a telling sign that there might be an issue – and this must have bearings on the Mo isotope composition of these reservoirs. At least, the authors must provide strong arguments (not just a good fit to a normal distribution in histograms) to assume otherwise. Just as a side note, trace elements concentrations in nature cannot be normally distributed in the first place.

Line 192: I would argue that "robustly" is quite a bold statement given that there are some open questions to be addressed.

Lines 203 to 205: However, by biasing the lithologies towards "intrusions", not only the Mo concentration data may be compromised (also see Fig. 2) but also the Mo isotopic composition. It is briefly mentioned in lines 207 to 209 – but not discussed further.

I presume the authors overall argument strategy is that given all their different estimates/models converge somewhat at a common value this, in retrospect, validates their approach?

Line 210: Isn't it the case that the uncertainty given here (<0.02 permil) is based on average compositions of different crustal

reservoirs rather than all of the individual samples? As such it is much smaller than the true variation observed for the whole dataset. Does it make sense to compare this value with the average analytical uncertainty?

Lines 236 and 237: Using the numbers given here, the oceanic crust is about 30% the volume of the continental crust. How justified is it to call that “significantly smaller”?

Lines 282 to 292: While the numbers might add up, inevitably, the models still rely on a rather limited dataset and thus, contain a considerable level of uncertainty. I would presume that such projections over Earth’s entire geological past should be considered as hypothetical (i.e. in the context of testable by other means) rather than as an outcome.

Version 1:

Reviewer comments:

Reviewer #2

(Remarks to the Author)

I have read the revised manuscript as well as the reply of the authors to my comments on the initial version of the manuscript. I think it is now generally in a very good state and that the issues I had raised have been adequately addressed.

Response to Reviewer #1:

This manuscript 'Missing molybdenum and the composition of the continental crust inferred from molybdenum isotopes' by Tian et al. have measured Mo isotope data in rocks from Gangdese arc which preserves a continuous section of lower-middle-upper crustal rocks. The authors have used their data combined with further data compilation to come up with Mo concentration and Mo isotope ratio of lower, middle, upper crust and bulk continental crust. This work is very timely and has great relevance to the field of research in stable isotope geochemistry, crust-mantle evolution.

Thank you for your support and recognition of the timeliness and scientific rigor of our work, as well as for your valuable feedback.

However, in the current state, the manuscript has some critical shortcoming which must be rectified before the manuscript is published. The authors have only measured the rocks from the middle crustal section of the arc and there is significant potential if rocks from upper and lower crustal section are investigated and are incorporated in this study.

We have made targeted improvements to our manuscript. Building upon our existing 13 middle continental crust (MCC) samples, we have added 5 upper continental crust (UCC) and 6 lower continental crust (LCC) samples from the Gangdese arc crustal section to facilitate more robust understanding. Additionally, we noted that a recent study has conducted preliminary research on the Mo isotopic composition of lower crustal rocks in the Gangdese Arc (18 data, Sun et al., 2025, *Geochimica et Cosmochimica Acta*, 398, 152–162). We have now included these data in our compiled dataset to further substantiate our conclusions.

Two major concerns that I have from this manuscript are presentation of isotopic data and calculation of main input parameters controlling the isotopic values of different crustal sections. Basic details about isotope measurements and measured values of standards are missing which hinders in assessing the quality of the data. In the present form, the work cannot be reproduced. The authors are requested to look at the line comments where it is mentioned in detail about why some of the input parameters are likely erroneous, particularly the $\delta^{98/95}\text{Mo}$ of the UCC (Upper Continental Crust). The Significant changes are therefore needed which will require re-running the models, which may change the proposed value of the BCC (Bulk Continental Crust).

In the supplementary Text section, we have added detailed analytical methods for Mo isotope geochemistry and the approach used to assess the Mo isotopic composition of the crust. This validates the accuracy of our inferences regarding the Mo isotopic composition of the continental crust. In Supplementary Table 2, we have also included the Mo isotopic compositions of the rock reference materials AGV-2 and W-2a as well as IAPSO Atlantic seawater, obtained during our analytical work. These results are consistent with previously published Mo isotopic compositions within analytical uncertainties, demonstrating the reliability of our data.

Based on your detailed line-specific comments, we have meticulously revised the entire manuscript. All modifications made in the manuscript and appendices have been highlighted in yellow for easy identification. Our work accurately reveals the Mo isotopic composition of different layers of the continental crust, as well as that of the bulk continental crust, and we believe our present study will provide new insights for future researchers.

Line Comments:

Line 42 - 43: Significant amount of work using Mo isotopes has been done in the recent years on different settings to understand the crust mantle evolution through geological time. Do also refer to such more recent works from the past 2-3 years, which cover these aspects of crust-mantle evolution and interaction to strengthen the point, that is, better constraints are needed for Mo isotope composition of different geochemical reservoirs.

We have thoroughly revised the introduction to emphasize that characterizing the geochemical compositions of Earth's distinct interior reservoirs serves as a fundamental cornerstone for deciphering interactions between end-member components and material cycling processes. Additionally, we have incorporated key recent literature (past two years) to strengthen our arguments, including: Chen et al. (2025, *Earth and Planetary Science Letters*, 658, 119294); Stegner et al. (2025, *Geochimica et Cosmochimica Acta*, 388, 294-306); Zeng et al. (2025, *Geology*, 53, 343-348); Fan et al. (2025, *Chemical Geology*, 680, 122683); Zhao et al. (2024, *GSA Bulletin*, 137,1855-1871). (Lines 51-54)

Line 76 - 78: I was unable to find the details of Mo isotope measurements in the manuscript and in the supplement. The authors have only reported the measured Mo isotope ratios. Only with this

information it is hard to assess the quality of the data as no sample preparation, column chemistry and measurement protocol has been given. There is no mention of Ru correction and procedural blank. There is also no report of the isotope spike, solution and whole rock standard(s) that have been used (if so) during the measurement. Without the measured Mo concentration and Mo isotope data of the solution and whole rock standards, and details of the measurement procedure, one cannot assess the quality of the Mo isotope data of the samples reported here. Therefore, the authors must include a section dedicated to the details of Mo isotope measurement.

We regret this oversight in the original manuscript. In the revised Supplementary Table S2, we have included data for reference materials AGV-2, W-2a, and IAPSO Atlantic seawater that were analyzed concurrently with our samples. Their Mo isotope compositions show excellent agreement with published values within analytical uncertainty (Chen et al., 2019, *Nature Communications*; Willbold et al., 2016, *Geostandards and Geoanalytical Research*; Zhao et al., 2016, *Geostandards and Geoanalytical Research*), confirming the reliability of our results. For isotopic analysis, we employed a ^{97}Mo - ^{100}Mo double spike, which was originally developed for Mo isotope measurements by Siebert et al. (2001, *Geochemistry, Geophysics, Geosystems*) and later validated for analytical accuracy by Zhang et al. (2018, *Journal of Analytical Atomic Spectrometry*). During the measurement, the total procedural blanks for Mo were 0.39 ± 0.37 ng (2 s.d.; n = 7). It is worth noting Ru was not detected in the analyte solution by ICP-MS in the separated Mo solution. Therefore, after this chemical procedure, no Ru correction is required for sample analysis. Additionally, during MC-ICP-MS measurements, we monitored the ^{99}Ru signal using the H3 Faraday cup, which was also negligible. Accordingly, we have expanded the Methods section to provide a detailed description of our Mo isotope analytical procedures (Supplementary Text Analytical methods; Lines 87-161).

Line 78: The authors need to explain the mineralogy of the studied samples to better classify them and rule out alteration. Even basic REE diagrams have not been shown. Given the implications the authors are trying to derive from these few samples, much detailed discussion is needed about them. In response, we have supplemented detailed mineralogical descriptions of the studied lithologies in the supplementary Text (Geological setting and samples of Gangdese arc crustal section and Fig. S2). These samples exhibit petrographic characteristics consistent with the previous studies in the

area (e.g., Guo et al., 2020, *Contributions to Mineralogy and Petrology*, 175, 58), showing fresh mineral surfaces devoid of significant alteration features. A new figure that depicts the REE patterns Gangdese arc magmatic section samples has been added (Fig. S3), which further demonstrates the pristine nature of these rocks.

Furthermore, we have classified the rocks based on SiO₂ content and provided new chondrite-normalized REE patterns and primitive mantle-normalized multi-element diagrams in Supplementary Fig. S3. All lithologies display coherent trace element signatures, confirming that: (1) they faithfully represent distinct crustal levels (UCC, MCC and LCC), and (2) their chemical compositions remain unaffected by post-magmatic low-temperature metamorphism or weathering processes. (Lines 91-96)

Line 79: The supplementary table S1 that is mentioned here has whole-rock data. If the whole rock data was measured by the authors as part of this study, details of how the whole rock data was measured must be given. This is one more instance where basic information of data gathering is totally absent. Also, the isotope data (Sr, Nd) that has been reported about these samples, were they measured by the authors? If so, then details of those measurements also must be given. If not, then the source of these data must be cited.

Even though the authors mention the Sr and Nd data in table, this data has not been used anywhere in the manuscript, and I seem to not find any discussion related to this data. It might be helpful for the authors to discuss the Sr and Nd isotope data in the context of mid crustal evolution.

In this revised version, we have made the following improvements: (1) All whole-rock major and trace element data in Supplementary Table S1 are our original analytical results, and we have added detailed analytical methods in the Supplementary Text; (2) Our results are consistent with previous studies of Late Cretaceous Gangdese arc magmatic rocks that exhibit uniform Sr-Nd isotopic compositions characteristic of juvenile lower crust (such as Guo et al., 2020), as shown in our initial data; (3) Since the main objective of this study is to reveal the Mo isotopic composition of different arc crustal levels (which can be determined by whole-rock geochemical characteristics alone, whereas Sr-Nd isotopes cannot distinguish crustal levels), we have removed the Sr-Nd isotopic data in this revision to better focus on crustal stratification and Mo isotopic characteristics. This modification allows for more targeted discussion of the key scientific issues.

Line 125 -187 In this portion of the manuscript the authors have gone into details about the possible Mo concentration and $\delta^{98/95}\text{Mo}$ of the UCC, MCC and LCC. These are the fundamental input parameters for the newly proposed isotope composition of BCC. However, the way they have chosen the [Mo] and $\delta^{98/95}\text{Mo}$ raises significant questions. The authors have indicated the effect of the hydrothermal fluid activity on Mo isotope fractionation and have mentioned they have taken steps to exclude samples with ore-bearing fluids. If we look at the citations, they have used to calculate the [Mo] and $\delta^{98/95}\text{Mo}$ of UCC, all of them (Xue et al, Shen et al., Kaufmann et al., Fan et al., Greber et al.) except one (Yang et al and some fresh igneous rocks in Greber et al.) are rocks with hydrothermal alteration and/or ore-fluid related process. Also bracketing rocks from same hydrothermally altered/ore-fluid related processes based on their SiO_2 content to represent different portions of the crust completely ignores the local geological effects on the [Mo] and $\delta^{98/95}\text{Mo}$. It is therefore obvious that the [Mo] and $\delta^{98/95}\text{Mo}$ derived from these rocks will be erroneous if implemented for UCC and in some cases MCC and LCC, a conclusion they themselves have come to later for [Mo]. But surprisingly even though the authors accept the [Mo] calculated by them through the compilation is missing [Mo], they have accepted the calculated $\delta^{98/95}\text{Mo}$ from the same samples without any deliberation. $\delta^{98/95}\text{Mo}$ of UCC that the authors are using in the model is $0.000 \pm 0.053\%$, which significantly differs from any previous estimates and the value commonly used (around 0.15; McCoy-West et al. and references therein). The input parameters for the [Mo] and $\delta^{98/95}\text{Mo}$ need to be changed after careful consideration and the subsequent models must be adjusted accordingly, to give new estimates on BCC $\delta^{98/95}\text{Mo}$. Another aspect of this study which can significantly improve and should be implemented in the manuscript is measuring the [Mo] and $\delta^{98/95}\text{Mo}$ of the rocks that represent the UCC and LCC portions of the Gangdese arc. This will give the authors [Mo] and $\delta^{98/95}\text{Mo}$ of rocks which are from different portions of the crust, which will significantly diminish the need of using $\delta^{98/95}\text{Mo}$ from compiled rock data which may not actually represent the different crustal sections. The Mo isotope data from surface sediments (sediments being a good representation of the exposed crust) should also be considered during estimation of the UCC composition.

We sincerely appreciate your valuable comments and suggestions. Existing studies have revealed distinct layered structures in the continental crust, with geophysical and geochemical (e.g. REE) evidence showing characteristic differences at various depths. This study therefore aims to

comprehensively estimate the Mo content and isotopic composition of upper, middle, and lower continental crust by compiling and integrating all currently published magmatic rock data. Following this approach, we did not specifically differentiate rock types from different regions or their potential mineralization processes, but focused on removing samples that have been affected by alteration as this has been clearly shown to affect Mo isotopic composition.

In response to your suggestions, we have strengthened our targeted investigation of upper, middle and lower crustal samples from the Gangdese arc. In this revised version, we have added 5 felsic samples representing the upper crust and 6 ultramafic-mafic samples representing the lower crust of the Gangdese arc, which is supplemented with recently published Mo isotope data for the Gangdese lower crust (18 data, Sun et al., 2025, *Geochimica et Cosmochimica Acta*, 398, 152-162). Our results demonstrate the Gangdese upper crust indeed exhibits lighter Mo isotopes ($\delta^{98/95}\text{Mo} = -0.17$ to -0.09% , Table S2) than previously published estimates for the UCC, which are closer to our newly estimated average for the UCC and may significantly advance current understanding of UCC Mo isotopic composition. (Lines 87-91)

To ensure robust results, we have compiled the largest dataset to date for estimating crustal Mo isotopes, now including 330 samples (159 samples for intrusive rocks with $\text{SiO}_2 \geq 63\%$, 52 samples for $52\% < \text{SiO}_2 < 63\%$, and 119 samples for $\text{SiO}_2 \leq 52\%$; Figs. S6-7). While we acknowledge that hydrothermal/fluid activities during magmatism may alter Mo isotopes, this is a subordinate process and will only effect rare samples. Instead, we have cross-validated our results with surface sediments (diamictites, clays and loess). Our compiled $\delta^{98/95}\text{Mo}$ values for these materials (diamictites, clay, and loess) yield a median of -0.070% (Fig. S8; Table S4, Supplementary Text), which is remarkably consistent with that of felsic intrusive rocks. This agreement strongly supports the validity of using felsic intrusive rocks to represent the Mo isotope composition of the upper continental crust. Finally, after careful consideration, we have refined the input parameters for both Mo concentrations and $\delta^{98/95}\text{Mo}$ values (Table S3, S4). These adjustments have led to updated estimates for the bulk continental crust (BCC) $\delta^{98/95}\text{Mo}$ composition of $-0.116 \pm 0.011\%$ (2 s.d.; $n = 6$; Fig. 3)

Line 217: Model presented here must be readjusted according to the modified BCC value.

In the revised manuscript, we have recalculated the bulk crust $\delta^{98/95}\text{Mo}$ values (Fig. 3) and

reassessed the crust-mantle evolution processes based on the newly determined crustal Mo isotope composition. (Lines 253-328)

Response to Reviewer #2:

First, the role of sediments and ore-related rocks appears to be not properly acknowledged in this study. At several points, the author use “converging values” as an argument for validating their results. For instance, they argue that a perceived normal distribution of Mo concentrations in intrusive crustal rocks justifies their choice of specific average Mo concentrations for the lower, middle and upper continental crust, although trace element concentrations in rocks cannot be normally distributed (e.g. also see Fig. S4, that would suggest negative Mo concentrations in some crustal rocks if this would be the case).

We sincerely appreciate your valuable comments and suggestions. The primary objective of this study is to estimate the Mo isotopic composition of the continental crust by using samples with different SiO₂ contents as proxies for upper, middle and lower crustal compositions, based on their distinct geochemical characteristics. With a dataset of this size, it is not necessary to evaluate the mineralization process for each individual sample, within the samples representing the UCC, mineralized outliers do occur but these do not constitute the dominant portion of the samples. In addition, the upper crustal sediments exhibit Mo isotopic compositions consistent with those estimated from the intrusive rocks in this study (see below). These independent datasets demonstrate that the potential inclusion of a small number of mineralized samples would not significantly affect our conclusions.

In our initial manuscript, we employed three methods to determine Mo concentrations for different crustal depths: arithmetic mean, median, and the "converging value" (highest probability value derived from Gaussian distribution) (Table S3). While the arithmetic mean showed higher values potentially due to outlier effects, the other two methods yielded nearly identical Mo concentration estimates across crustal layers. As you pointed out, trace element concentrations in rocks do not follow normal distributions. Therefore, in the revised version we have adopted instead the median values as more representative estimates of Mo concentrations for different crustal depths.

Also, they somewhat dismiss the effect of sediments and ore-related rocks on the Mo isotope budget of the continental crust, arguing that because different models converge at a similar value somewhat verifies this view.

We have incorporated an evaluation of sedimentary rocks (diamictites, clay, and loess) in the

Supplementary Text. These sediments are widely recognized as excellent proxies for UCC composition. Our compilation shows they yield a median $\delta^{98/95}\text{Mo}$ of -0.070‰ (Fig. S8; Table S4, Supplementary Text), which is remarkably consistent with felsic intrusive rocks. This agreement between completely independent sedimentary records and intrusive rocks strongly validates our approach of using intrusive compositions to estimate UCC Mo isotopic signatures. Though ore-forming fluids may locally modify Mo isotopic compositions in rocks, their crustal-scale influence appears limited. This study employs intrusive rocks with distinct seismic wave velocities to represent different crustal levels, effectively filtering out mineralization-related anomalies through systematic sampling.

At last, the fact that this study is based on merely 250 data points for a reservoir that shows the largest isotopic and geochemical variations of all geochemical reservoirs puts a large uncertainty on their conclusions. In this context, it does not help that the authors freely switch between using 2 sigma and 2 sigma mean uncertainty estimates in their models and conclusions.

In the revision, we have significantly expanded our dataset by adding new Gangdese arc magmatic samples: 5 felsic samples representing upper crust, 6 ultramafic-mafic samples representing lower crust, combined with our existing 13 intermediate samples (middle crust) and 18 recently published lower crustal samples from the Gangdese arc (Sun et al., 2025, *Geochimica et Cosmochimica Acta*, 398, 152-162). This has allowed us to establish distinct Mo isotopic signatures at different crustal depths within the Gangdese arc. Additionally, we have now compiled the largest dataset to date for estimating crustal Mo isotopes, totaling 330 samples (159 intrusive rocks with $\text{SiO}_2 \geq 63\%$, 52 samples with $52\% < \text{SiO}_2 < 63\%$, and 119 samples with $\text{SiO}_2 \leq 52\%$; Figs. S7-8). The results demonstrate that the Gangdese arc rocks, are similar to globally representative samples from different continental crustal levels, exhibit distinct Mo isotopic compositions corresponding to different crustal depths. We have included all available published data in our modelling making our conclusions as robust as possible.

To ensure data reliability, all Mo elemental and isotopic data are presented with both 2 s.d. and 2 s.e. uncertainties, using consistent reporting standards in data tables. The results continue to show clearly distinct Mo isotopic compositions at different crustal depths, providing the foundation for our improved estimate of BCC Mo isotopic composition.

Introduction

This provides an historic overview of the development of the Mo isotope system to target Earth Science-related questions. This has long been the standard way of introducing the system in publications, i.e. starting with low-temperature applications and ore geology before homing in on their application to magmatic processes. I find this quite old-fashioned given that there is now ample literature on the application of Mo isotope to geodynamic questions. I would suggest the authors re-work this part accordingly.

The introduction section has been revised accordingly, we have first elaborated on the fundamental significance of continental crust studies, then addressed key unresolved issues in crust-mantle interaction dynamics, before highlighting the unique value of Mo isotopes in deciphering these Earth system processes. In particular, we emphasized that the Mo isotopic composition of the continental crust remains a critical open question - which further underscores both the importance and urgency of our current research. (Lines 38-79)

Line 52: I do not agree with this notion. As indicated by the authors, the uncertainty of the average composition of chondrites is small (accordingly given as 2 standard error) but this does not mean that the Mo isotopic composition of the chondritic reservoir is homogeneous. Not appreciating this here, I fear that this is (already) a major obstacle in what will follow in the context of the overall aim of this study.

We agree with you that the chondritic reservoir includes heterogeneities. However, the currently accepted average $\delta^{98/95}\text{Mo}$ value of chondrites is used in numerous studies (e. g., McCoy-West et al., 2019, *Nature Geoscience*; Greber et al., 2015, *Earth and Planetary Science Letters*; Hin et al., 2022, *Earth and Planetary Science Letters*; Willbold and Elliott, 2017, *Chemical Geology*) and is our best available approximation of the building blocks of Earth. This type of approach has been used in geochemical studies looking at coupled evolution of the crust-mantle system for > 50 years and therefore should not be a reason to question the robustness of this study in particular.

Line 57: Please provide context. Unchanged compared to what?

We have incorporated the comparative reference (compared to previous estimates based on either

granites or subduction-related volcanic rocks³¹⁻³²) in this sentence. (Line 69)

Line 63: This is misleading. Please rephrase. I presume the authors mean that the estimated d98Mo value for bulk CC is biased towards upper CC values due to sampling bias, not that the databases for $\delta^{98/95}\text{Mo}$ in upper CC rocks are biased (e.g. because of analytical artefacts)?

We have rewritten the sentence as “*current estimates of the bulk continental crust (BCC) Mo isotopic composition are derived through extrapolation from upper crustal samples that dominate existing databases*”. (Lines 73-75)

Lines 72 to 80: Even though the Gangdese arc might have been recently discovered, I believe that this is more of a regional interest. Quite on the contrary, I think that adding the 13 data points for the samples from the Gangdese arc (here and in the next section – i.e. lines 87 to 187) in this context is misleading. This small number of data points do provide some additional information but, truly, not much and the whole project is built on literature data in the first place. As such, I would re-write these parts of the manuscript with a focus on the literature data only and, accordingly, move focus away from the (small number) of additional new data.

In this study, we propose to use the Gangdese arc crust as a reference framework to initiate the discussion on continental crust composition, with the ultimate goal of constraining the Mo isotopic composition of the bulk continental crust. The revised manuscript presents new Mo isotope data for upper, middle, and lower crustal rocks from the Gangdese arc, which are then integrated with compiled global datasets to provide more robust constraints on the overall continental crust composition. (Lines 87-98)

Line 119: Using a highly incompatible/highly compatible element pair (such as Rb/Ni) is an odd ratio. What is the underlying reason for this? Why not use Eu/Eu*, which would be the obvious choice when discriminating between the UCC and the LCC?

Although the Rb/Ni ratio is not commonly utilized, it exhibits distinctive characteristics in discriminating samples from different crustal levels. Given that Rb behaves as an incompatible element while Ni is compatible, Rb becomes the dominant variable during upper-middle crust evolution while Ni remains relatively constant, with the opposite trend observed in the lower crust.

Consequently, when combined with SiO₂ content, this ratio can effectively reveal a sample's crustal affinity.

In contrast, the widely used Eu/Eu* ratio is strongly controlled by plagioclase, which may persist in or fractionate from the melt during partial melting across all crustal levels (lower, middle, and upper crust). Moreover, the Eu/Eu* ratio may also be influenced by the oxygen fugacity (fO_2) during rock formation. These inherent variabilities render simple Eu/Eu*-based discrimination of crustal depth unreliable.

Lines 120 to 125: I don't think one can use the trace element systematic for individual crustal samples to attribute them to a specific position within the crust ("... confirming their formation in the MCC"). I find this a very long stretch. Please provide further substantiation.

We fully agree with your view that trace element characteristics alone cannot reliably determine the specific crustal depths of rock samples. In this study, our ultramafic-mafic, intermediate, and felsic samples were systematically collected from the well-characterized Gangdese crustal transect (Guo et al., 2020, *Contributions to Mineralogy and Petrology*, 175, 58), where previous seismic velocity measurements have unambiguously confirmed their correspondence to lower, middle, and upper crustal layers, respectively.

To further verify the crustal affiliations of our samples, we have cross-referenced established compositional ranges (SiO₂ content and trace element ratios) for different crustal depths from previous studies (Fig. S5). The excellent consistency between our samples and these reference values provides additional confidence in their origins within the stratified crustal section.

In response to your suggestions, we have streamlined the related descriptions in the manuscript to improve its overall flow while maintaining the essential evidence supporting our sample classification. (Lines 122-128 and 139-141)

Lines 135 to 149: I am still not convinced that this simple "chemo-stratigraphic" approach is valid. However, there is an easy test that I would ask the authors to carry out in a revised version. If the Mo concentration of LCC, MCC, and UCC can be estimated by querying the SiO₂ content of analysed rocks, then the identical approach should work with other incompatible elements. This means that taking the same samples and applying the SiO₂ filter, the so-calculated average

concentrations for all other incompatible trace elements should match the estimates of the same elements in the LCC, MCC and UCC in the literature (e.g. Rudnick and Gao).

This is a particularly intriguing question. While current models based on the SiO₂ content of crustal rocks show good correlation with seismic wave velocities and crustal depth—as demonstrated by Xia et al. (*Geology*, 2025, 53, 18-22), who established that ultramafic, mafic-intermediate, and felsic rocks correspond to the lower, middle, and upper continental crust—it remains an open question whether trace element concentrations in these rock types can be strictly equated with the established compositional profiles of different crustal depths.

In fact, our study reveals significant discrepancies in Mo concentrations between the compiled data of rocks with varying SiO₂ contents (representing different crustal depths) and the estimates from Rudnick and Gao (2014) (Fig. 2). Such variations are not uncommon, as different studies often report divergent estimates for specific element concentrations at given crustal levels (e. g., Gao et al., 1992, *Geochimica et Cosmochimica Acta*; Sammon and McDonough, 2021, *Journal of Geophysical Research: Solid Earth*). This makes it challenging to validate the accuracy of results solely by comparing trace element compositions of specific SiO₂-defined samples with a single existing model.

For our current research, however, several lines of evidence support the robustness of our approach: (1) The Mo isotopic composition of felsic intrusive rocks aligns closely with that of crustal sediments, strongly validating our estimates for the upper crust (Table S4); (2) Lower crustal xenoliths and exposed ultramafic rocks consistently exhibit compositions expected for deep crustal materials (Rudnick and Gao, 2014); (3) The samples, when grouped by SiO₂ content, display seismic wave velocities matching their respective crustal depths (Guo et al., 2020; Xia et al., 2025). Thus, despite the complexities in trace element systematics, using rocks of distinct SiO₂ ranges to represent different crustal levels remains a methodologically sound approach. By using this significantly expanded compiled igneous dataset we are minimizing the large variability for Mo isotopes often observed in individual studies.

Line 152: Comma missing after “this”?

We have rewritten this sentence. (Line 171)

Lines 174 to 178: Is this used (later?) to validate the approach? If so, I would strongly disagree. Just because the data shows a normal distribution cannot be used to argue that the data is representative for the average composition of different crustal compositions. Something that “looks right” does not a priori imply it is right. If the authors think otherwise, they need to provide a comprehensive argumentation to do so.

I would argue that sedimentary rocks as well as ore-related lithologies are a very important lithology when estimating the bulk composition of crustal reservoirs. The fact that the authors approach underestimates the Mo concentrations in the LCC and UCC compared to established estimates from the literature is a telling sign that there might be an issue – and this must have bearings on the Mo isotope composition of these reservoirs. At least, the authors must provide strong arguments (not just a good fit to a normal distribution in histograms) to assume otherwise. Just as a side note, trace elements concentrations in nature cannot be normally distributed in the first place.

Indeed, while the normal distribution of data does not necessarily indicate representation of the average crustal composition, for our compiled samples, only those with the highest relative probability values can be used to represent the most probable average crustal composition—these values incidentally align with the normal distribution trend.

Sedimentary rocks at the Earth's surface are commonly used to represent the composition of the UCC. To evaluate this, we compiled published sedimentary rock data (including diamictites, clays, and loess). The results show that while their Mo concentrations vary widely (0.19–2.97 $\mu\text{g/g}$, median = 0.72 $\mu\text{g/g}$), they exhibit Mo isotopic compositions ($-0.070 \pm 0.119\%$; Fig. S8) consistent with our estimated upper crustal values. This further validates the robustness of our results.

Although ore-forming processes may locally modify the Mo isotopic composition of the crust—and indeed, some existing estimates of upper crustal Mo isotopic composition are derived from molybdenite (Greber et al., 2014, *Lithos*)—mineralization on a whole-crustal scale is a localized phenomenon. When using large-scale lithological data to represent different crustal layers, it is impractical to completely exclude such effects. However, with a large enough dataset these potential outliers caused by mineralization become inconsequential.

To clarify, in Supplementary Table S3, we have calculated three distinct estimates for Mo concentrations at different crustal depths: (1) Arithmetic average; (2) Median and (3) Highest probability value. Notably, the median and highest probability value exhibit nearly identical

characteristics. To prevent potential misinterpretations, we have adopted the median as the representative Mo concentration for each crustal depth in the revised manuscript.

Line 192: I would argue that “robustly” is quite a bold statement given that there are some open questions to be addressed.

We have removed the word “robustly”.

Lines 203 to 205: However, by biasing the lithologies towards “intrusions”, not only the Mo concentration data may be compromised (also see Fig. 2) but also the Mo isotopic composition. It is briefly mentioned in lines 207 to 209 – but not discussed further. I presume the authors overall argument strategy is that given all their different estimates/models converge somewhat at a common value this, in retrospect, validates their approach?

Our study demonstrates that while focusing solely on intrusive rocks may introduce some bias in Mo concentration (Fig. 2), the compiled Mo isotopic data from the UCC sedimentary proxies (including diamictites, clays, loess) exhibit remarkable consistency with our intrusive rock-derived UCC estimates (median $\delta^{98/95}\text{Mo} = -0.070\text{‰}$; Table S4). Therefore, we believe that biasing the lithologies towards “intrusions”, Mo isotope composition was not significantly compromised. In the other portions of the CC (i.e. MCC and LCC), intrusions are the dominant rock type and therefore it makes a lot of sense for those reservoirs.

Regarding the overall logic of the manuscript, we applied various estimates of Mo concentrations and different continental crust models to calculate the $\delta^{98/95}\text{Mo}$ of the BCC. Through evaluating the consistency of the results and the reasonableness of the input parameters, we derived an estimate for the $\delta^{98/95}\text{Mo}$ of the BCC. Finally, we employed a model involving a complementary relationship between the crust and the depleted mantle to further validate the plausibility of the BCC estimate and to infer the history of crustal extraction.

Line 210: Isn't it the case that the uncertainty given here (<0.02 permil) is based on average compositions of different crustal reservoirs rather than all of the individual samples? As such it is much smaller than the true variation observed for the whole dataset. Does it make sense to compare this value with the average analytical uncertainty?

The uncertainty given here ($< 0.02\%$) is based on using the same Mo concentrations and $\delta^{98/95}\text{Mo}$ for the LCC, MCC, and UCC (Fig. 3), but utilizing the different crustal depth models, the maximum variation in the $\delta^{98/95}\text{Mo}$ of the BCC is $< 0.02\%$. As this value is smaller than the average analytical uncertainty of the BCC, it suggests that the use of different crustal models to estimate $\delta^{98/95}\text{Mo}$ of the BCC does not significantly influence the result within the uncertainty of the approach.

Lines 236 and 237: Using the numbers given here, the oceanic crust is about 30% the volume of the continental crust. How justified is it to call that “significantly smaller”?

We have deleted ‘significantly’ here.

Lines 282 to 292: While the numbers might add up, inevitably, the models still rely on a rather limited dataset and thus, contain a considerable level of uncertainty. I would presume that such projections over Earth’s entire geological past should be considered as hypothetical (i.e. in the context of testable by other means) rather than as an outcome.

We fully acknowledge your perspective regarding the anticipated increase in crustal Mo isotope data in future studies. All models are only as valid as the inputs. However, while expanded datasets may refine the estimated Mo isotopic composition of the crust, they would not fundamentally challenge this study's conclusions about crust-mantle differentiation and evolution (in this revision the database was expanded from 250 to 330 values; incorporating 32% more data, however, the final BCC composition is essential unchanged). In fact, the work by McCoy-West et al. (2019, *Nature Geoscience*, 12, 946–951) has demonstrated that crust-mantle system likely reached a balanced state as early as 3.5 billion years ago, persisting through the early Cenozoic. Our study further substantiates this view with comprehensive Mo isotopic data across the entire crustal column and confirming the complimentary between the crust and depleted mantle.

Response to Reviewer #2:

I have read the revised manuscript as well as the reply of the authors to my comments on the initial version of the manuscript. I think it is now generally in a very good state and that the issues I had raised have been adequately addressed.

Thank you for your approval of the revised version of our manuscript.